# Comparison of Storage-Related Volatile Profiles and Sensory Properties of Cookies Containing Xylitol or Sucrose

**DOI:** 10.3390/foods12234270

**Published:** 2023-11-26

**Authors:** Jaroslawa Rutkowska, Damian Baranowski, Agata Antoniewska-Krzeska, Eliza Kostyra

**Affiliations:** Institute of Human Nutrition Sciences, Faculty of Human Nutrition, Warsaw University of Life Sciences, Nowoursynowska St. 159c, 02-776 Warsaw, Poland; damian_baranowski@sggw.edu.pl (D.B.); agata_antoniewska@sggw.edu.pl (A.A.-K.); eliza_kostyra@sggw.edu.pl (E.K.)

**Keywords:** alternative sweetener, xylitol, cookies, volatile compounds, sensory attributes, storage

## Abstract

Excessive consumption of simple sugars is responsible for non-communicable diseases such as obesity, cardiovascular diseases, and diabetes. Xylitol has anticarcinogenic, prebiotic-like characteristics and a lower glycaemic index and caloric value than sugars, which makes it a valuable alternative sweetener. The aim of this study was to examine the effects of storage of volatile compounds and sensory profiles of cookies containing xylitol as a sucrose alternative or sucrose by applying solid-phase microextraction gas chromatography/mass spectrometry and quantitative descriptive analysis. The volatile compound profiles of both kinds of cookies were similar, especially regarding markers of Maillard reactions (Strecker aldehydes, pyrazines) and unfavourable compounds (aldehydes, hydrocarbons, and organic acids). Throughout the period of storage lasting 0–9 months, the total content of hydrocarbons was stable and averaged 10.2% in xylitol cookies and 12.8% in sucrose cookies; their storage for 12 months significantly (*p* < 0.05) increased the contents to 58.2% and 60.35%, respectively. Unlike sucrose, xylitol improved the stability of the pH and water activity of cookies and sensory attributes such as buttery aroma and texture characteristics during 12 months of storage. The results indicated that 9 months of cookie storage was the maximum recommended period. The inclusion of xylitol in cookies might replace sucrose and high-fructose-corn syrup and synthetic additives commonly used in industrial production.

## 1. Introduction

Sweet bakery products (cookies, pies, pastries, and muffins) are known as the main source of simple sugars in the human diet. Sucrose is one of the principal sweeteners used in cookies. High-fructose-corn syrup (HFCS) is commonly used as an added sweetener substituting for sucrose in industrially produced pastries because of its low cost and beneficial functional properties. HFCS that has 42% or 55% fructose possesses a relative sweetness similar to that of sucrose (90% and 99%), imparts a suitable aroma and colour and good stability of pastries, and is distinguished by ease of supply in comparison with granulated sucrose. However, because of its functional properties, e.g., providing lubricity, flavour, and mouthfeel, sucrose is still desired in the manufacturing of cookies, biscuits, and cakes [1,2,3].

Excessive consumption of sugar dramatically increases the number of people suffering from obesity, cardiovascular diseases, and diabetes [2]. Almost 60% of adults in Europe are affected by overweight, including nearly 25% of obese adults. A similar pattern also applies to children (33% and 10%, respectively) [4]. The leading demographic of sweetened beverage and snack consumers is adolescents, who may develop neurodevelopmental changes persisting into adulthood as a result of this overconsumption. Adolescent HFCS consumption leads to protracted dysfunction in affective behaviours and alterations in accumbal proteins, which persist even after cessation of HFCS consumption [5].

Even short-term consumption of HFCS containing 55% fructose by animals aggravated colitis and upregulated the proportion of macrophages in mice with inflammatory bowel disease [6]. A high-glucose or high-fructose diet increased intestinal permeability, dysregulated the gut microbiota, and changed tight junction proteins [7].

Sugar alcohols (polyols) can be considered more appropriate sweeteners because of their potential health benefits. Compared with sucrose and HFCS, polyols are poorly absorbed and therefore provide fewer calories and lower glycaemic responses, as indicated by the glycaemic index (GI) of erythritol (0), xylitol (13), or maltitol (35), in contrast to sucrose or HFCS (56–65) [8,9].

Among polyols, xylitol is equisweet to sucrose and is best known for its dental benefits, such as reducing the risk of dental caries by reducing the growth of pathogenic *Streptococus mutants* [10]. In 2008, the European Food Safety Agency approved a health claim “xylitol chewing gum reduces the risk of caries in children” [10,11]. Due to its prebiotic-like characteristics, xylitol deserves particular attention [12]. In the digestive tract, xylitol undergoes about 50% absorption, and the rest enters the colon, where it can serve as an energy source for the intestinal probiotic microbiota [10]. An in vitro study revealed the prebiotic potential of xylitol and sorbose in promoting the growth and metabolic activity of specific butyrate-producing bacteria—*Anaerostipes* spp. in human faecal culture [13]. The authors found that xylitol stimulated the formation of butyrate (˃20 mM) to a higher degree than did the representative prebiotics: fructooligosaccharide (FOS) and galactooligosaccharide (GOS) [13]. Other researchers found that xylitol promoted the growth of beneficial bacteria, such as *Bifidobacterium* and *Lactobacillus* in the rat colon [12]. Also, a more recent animal (mouse) study showed that consumption of 5% xylitol in diet (0.44 g/kg/day for human beings) increased the relative abundance of *Bifidobacterium* and *Lactobacillus*, maintained *Lachnospiraceae*, and decreased the relative abundance of *Escherichia* and *Staphylococcus* [12]. The authors concluded that the key enzymes for xylitol digestion from different bacteria can support the growth of microbiota and also enhance the concentration of propionate, which lowers pH to restrict the growth of pathogenic *Escherichia* [12]. Further research by Zhang et al. [14] suggests that xylitol and mannitol may improve the intestinal flora of diabetic patients, increase probiotics and play a role in hypoglycaemia, but their specific mechanisms still need to be further studied. The study also showed that xylitol could prevent other diseases, for example, some respiratory diseases, e.g., pneumonia and middle ear infection, and efficiently stimulate the immune system and lipid and bone metabolism [15,16]. Xylitol may thus be regarded as a recommended sweetener for those suffering from diabetes, cardiovascular disease, and obesity, and has the added benefit of maintaining colon health, as it favours saccharolytic anaerobes and purifies the colon of endotoxic and pathological organisms [9,10,15].

Previous studies on the replacement of sucrose by polyols in bakery products and cookies (also in gluten-free products) included assessments of the rheological properties of dough, textural, physical, and sensory properties, and the acceptability of baked products [1,17,18,19,20]. The results indicated that maltitol, isomalt, and sorbitol may be suitable as sucrose replacers in muffins as they positively affected the instrumental texture and consumer acceptability [20]. Other studies revealed that the replacement of sucrose with polyols significantly affected sensory characteristics. The surface colour and sweetness of Danish cookies were scored lighter, as the erythritol level increased and the sucrose level decreased [18]. The sweetness of cookies with xylitol was rated the same as those with sucrose, and it showed a pronounced cooling effect with some aftertaste [1,21]. The differences in sensory scoring sweetness probably resulted from various degrees of sweetness of erytritol (75%) and xylitol (100%) [17]. The shelf life of cookies containing xylitol was subjected to microbiological and sensory analyses, revealing a beneficial effect of polyols as functional ingredients that modulate processability and improve textural properties, such as those of sugar-sweetened bakery products [17,21]. Cookies containing xylitol instead of sucrose were sensually acceptable and microbiologically safe (during 14 days of storage) and tended to have a prolonged shelf life [21].

Volatile profiles of cakes containing sucrose and monosaccharides and partially replaced by sucrose alternatives (e.g., apple pomace, whey permeate, and oligofructose) or cookies containing various sugars (sucrose, fructose, and glucose) were reported [22,23]. Volatile compounds contained in pastries are derived mainly from Maillard reactions, caramelization, lipid oxidation, and thermal degradation [22,23,24]. The Maillard reaction, which occurs between amino compounds and reducing sugars, is one of the important routes to flavour and browning formation in baked confectionery products [24]. Aldehydes, which are considered important volatiles in pastries, can be classified into two groups. The first one refers to compounds originating from Maillard reactions, specifically arising from Strecker degradation of α-amino acid in the presence of α-dicarbonyls [25]. Volatiles from that group, such as 2-methylpropanal, 3-methylbutanal, benzaldehyde, and phenylacetaldehyde, are considered important contributors to the beneficial aroma attributes of baked confectionery products (almond, cherry, honey, floral, sweet, chocolate, and malty) [23]. Aldehydes belonging to the second group are mainly lipid oxidation markers (e.g., hexanal, octanal, nonanal, heptanal) and are responsible for such aroma attributes as fatty, oily, cake crust, damp, and musty [23,26,27]. Ketones are considered important contributors to the desirable aroma of pastries [25]. The decomposition of sugar during the baking process results in diketones, such as 2,3-butanedione (diacetyl) and 2,3-pentadione, which are responsible for favourable aromas in pastries: buttery, caramel, and butterscotch [23,25]. The volatile profile of cookies is also distinguished by the presence of methyl ketones (e.g., 2-butanone, 2 heptanone, 2-nonanone), contributing to buttery aroma [25,28]. Alcohols resulting from the oxidative degradation of unsaturated fatty acids are less numerous than aldehydes and ketones in the volatile profile of cookies. Most alcohols positively affected the aroma of pastries. However, in sucrose-containing sponge cakes, 1-hexanol and 1-octen-3-ol may be perceived as unpleasant at high concentrations [25,26]. Pyrazines are important volatiles formed in cakes with higher concentrations in the crust than in the crumb. In sucrose-containing sponge cakes, a range of pyrazine compounds were detected, e.g., methylpyrazine, 2,5-dimethylpyrazine, trimetylpyrazine, and 2-ethyl-pyrazine, which contributed to nutty, cake crust, and roasty aroma [25,26]. It should be noticed that the odour threshold values of methylpyrazines are relatively high (˃1 ppm); thus, they play an olfactory role at higher concentrations [29]. The low moisture content of biscuits and cookies favours the formation of furans during caramelization and Maillard reactions [25]. Volatiles from this group: furfural, 5-hydroxymethylfurfural (HMF), 2-furanmethanol, 2-acetylfuran, furaneol, and penthylfuran, which impart sweet, caramel, bready, spicy, and fatty aroma, revealed as important odour-active compounds in sucrose and monosaccharide- and sucrose-reduced cookies [22,23,25,26]. Furans and their derivatives have low odour thresholds, thus being probably important for perceiving the aroma of pastries [25].

In the analysis of volatile compounds in cookies and cakes containing sucrose, monosaccharides, and partially replaced by sucrose alternatives, gas chromatography with mass spectrometry (GC/MS) is used. The creation of mass spectra for each compound enables the identification of compounds in the sample by comparing library databases and retention indexes [25]. Regarding sample treatment, head-space extraction using solid-phase microextraction (SPME) has been often preferred because it is a rapid, simple, and solvent-free technique, and it allows to obtain a reasonable number of volatile compounds belonging to a wide range of chemical classes [22,23,26,30]. Volatile profiling is a useful tool that provides information about odour-active compounds that influence sensory attributes and for the detection of possible toxic compounds important with regard to consumer safety. However, the volatile profiles of bakery products containing polyols have not yet been studied.

Thus, the aim of the study was to examine the effects of xylitol as a sucrose alternative in cookies on generating volatile compounds and on the sensory profile during storage.

## 2. Materials and Methods

### 2.1. Preparation of Cookies

Cookies were prepared in six variants, differing in the type of sweetener (sucrose or xylitol) and its content (Table 1). Ingredients for cookie manufacturing were purchased from a local market in February 2022 in Warsaw, Poland: sucrose of 99.6% purity [Pfeifer & Langen Sp. z o.o., Poznań, Poland], butter containing 83% milk fat [Mlekpol, Grajewo, Poland], wheat flour, type 480, moisture content 13%, derived from Polskie Młyny, S.A. [Warsaw, Poland], and xylitol of 99.5% purity was produced from birch tree by Danisco Sweeteners [Kotka, Finland] and distributed by Sante [Warsaw, Poland].

The cookie preparation process consisted of mixing butter with powdered sweetener (sucrose or xylitol) and wheat flour containing 0.4 g/100 g ash for 5 min using a mixer, adding egg yolks, and mixing for 3 min again.

The dough was cooled in a refrigerator (1 h at 4 °C) and then cut into slices 5 mm thick, from which circular, 50 mm wide shapes were cut. Baking was carried out at 170 °C for 12 min in an electric oven (Hendi 225516, De Klomp, The Netherlands). After cooling (24 h from baking), the cookies were packed in ecological cellulose film (20 μm thickness, permeability to water vapor—20 g/m^2^·24 h, oxygen—5 cc/m^2^·24 h) packages, with eight cookies in each package. Next, packaged cookies were placed into cardboard boxes without light access. The packed cookies were kept under the same conditions at room temperature (18 ± 1 °C) and average humidity (60 ± 2%).

All formulations were prepared in duplicate. Baked cookies were assayed immediately after production and then after 3, 6, 9, and 12 months of storage (with regard to pH, water activity, volatile compounds, and sensory assessment). The microbiological analysis was performed at monthly intervals.

### 2.2. Physicochemical Properties (Water Activity Determination, pH Measurement) and Microbiological Analysis of Cookies

Water activity (a_w_) was determined in ground cookies using a Pre AquaLab water Activity Analyzer (Meter Group, Pullman, WA, USA) at 25 °C according to the procedure presented elsewhere [31]. The pH of cookie samples was measured in solution. Briefly, 10 g of ground cookie samples was placed in an Erlenmeyer flask containing 100 mL of ultrapure water (25 °C). Then, the content was mixed with an electronic agitator for 30 min. The pH was measured with an Elmetron CP-411 pH meter, equipped with the EPP-1 electrode [32]. The numbers of yeast and mold colonies were determined according to PN-ISO 21527-2 as presented elsewhere [33,34].

### 2.3. Analysis of the Volatile Compound Profile

Volatile compounds were extracted from cookies using headspace solid-phase microextraction (SPME) according to the reported procedure [34]. Solid-phase SPME was carried out by a fibre coated with a 50/30 mm thickness of DVB/CAR/PDMS (divinylbenzene/carboxen/polydimethylsiloxane; Supelco, Bellefonte, PA, USA). Before extraction, the fibre was conditioned by heating in a gas chromatograph injection port at 270 °C. For headspace SPME extraction, 5 g of ground sample was placed in a 20 mL vial, closed with a silicone-Teflon sealing cap, and heated at 40 °C for 30 min to stabilize the concentration of volatiles in the headspace. The sample was continuously shaken with a magnetic stirring bar.

The analyses were performed using gas chromatography/mass spectrometry (GC/MS) (6890 N GC, 5975 MS Agilent, Santa Clara, CA, USA). Volatile compounds were separated on a DB-624 capillary column (30 m × 0.25 mm i.d., 1.4 μm film thickness).

Volatile compounds were identified by comparing their mass spectra with those of the NIST.08 and Wiley 7th Ed (National Institute of Standards and Technology, Gaithersburg, MD, USA) libraries. For validation purposes, volatile compounds having only ≥80% similarity with the Wiley mass spectral library, were tentatively identified using GC-MS spectra. For the determination of linear retention indices (LRI), a mixture of n-alkanes (C6–C20) dissolved in n-hexane was employed (C6–C20, Sigma-Aldrich, Poznań, Poland) according to Van den Dool and Kratz [35]. The quantities of volatile compounds were expressed as relative peak areas (peak area of each compound/total area) × l00.

### 2.4. Sensory Evaluation of Cookies

Cookie samples were evaluated using quantitative descriptive analysis (QDA) [36,37]. The assessment was performed by ten panellistpanellists experienced in the evaluation of bakery products, prescreened for their sensory ability by basic taste and aroma detection tests and by their ability to describe and discriminate food products. The panellists, supervised by a moderator, selected the descriptors that best characterized the cookie samples. The final list comprised 18 sensory attributes (1 appearance, 8 aroma, 5 taste, and 4 texture). Attributes generated by the panel with their relevant definitions are presented in Appendix A.

The panellists evaluated the intensity of each attribute using a 10 cm linear scale in the following order: appearance (colour), aroma (buttery, sweet, caramel, vanilla, floral, nutty, roasty, fatty, floury, sour, and pungent), taste (buttery, sweet, roasty, bitter, and aftertaste—cooling effect) and texture (hardness, fracturability, moisture, and adhesiveness).

Samples were presented in transparent, disposable, plastic containers (150 mL) covered with lids, each containing one cookie. Containers were coded using random, 3-digit numbers. Six samples were simultaneously presented in a random order to each panellist at room temperature (21 ± 2 °C). To eliminate the residual flavour, the panellists received water. The assessments were made in a sensory laboratory room, fulfilling general requirements for sensory testing conditions according to ISO 8589 [38] in individual booths equipped with a computerized system with sensory software for data acquisition [Analsens NT, PAN, Poland]. The assessments of fresh and stored cookies were performed separately and made in two replications.

Before participating in the evaluation, the sensory panel read the information sheet and signed their informed consent. Our research protocol followed the guidelines of the Helsinki Declaration, and all procedures involving human subjects were approved by the Committee on Ethics in Human Beings Research of the Institute of Human Nutrition Sciences, Warsaw University of Life Sciences (Reference: 04/22).

### 2.5. Data Analysis

The results are expressed as the means ± standard deviations (SD). The data were subjected to multiway ANOVA followed by Tukey’s post hoc test using Statistica 13.3 (TIBCO Software Inc., Palo Alto, CA, USA). The level of *p* < 0.05 was considered significant. Principal component analysis (PCA) was performed for assessing the relationship between sweetener (type and content), storage, and sensory attributes from QDA estimation. A dendrogram was generated to visualise the clustering of the obtained data using XLSTAT (Lumivero, Denver, CO, USA).

## 3. Results and Discussion

### 3.1. pH Values, Water Activity, and Microbiological Assay

Cookies with xylitol had slightly higher (*p* < 0.05) pH values than those with sucrose (5.95 and 5.80, respectively; Table 2) but lower pH values than those reported for cookies and pastries by others (7.0–9.75). This might have been due to leavening additives: sodium, potassium, or ammonium bicarbonate, which were not used in this study [22,32,39]. Similar effects of sweeteners on pH values were reported by Cincotta et al. [22] for sucrose, fructose, and glucose (pH = 7.2, 6.4, and 6.6, respectively). Storage of cookies decreased pH values by approximately 0.3 only in those containing the highest amount of sucrose (23%). A much higher decrease in pH value (by 0.5–0.6) was reported for cookies stored for 15 months [40]. No effect of storage was found for cookies containing xylitol, which indicated their superior stability compared to those with sucrose (Table 2).

Fresh cookies containing xylitol had higher water activity (a_w_) values (0.352–0.386) than those with sucrose (0.327–0.367; *p* < 0.05; Table 2), as reported by Zoulias, Piknis, and Oreopoulou [19]. This may suggest that polyols bind free water in pastry and bakery products better than sucrose [19,41]. The higher the content of either xylitol or sucrose in the cookies was, the lower the a_w_ values, as also reported for xylitol by Sahin, Axel, Zannini, and Arendt [41], probably due to the high hygroscopicity of xylitol.

The water activity of cookies stored for 12 months was significantly increased (Table 2), as reported for stored cookies containing sucrose [42,43]. A lower increase in a_w_ was found for cookies containing xylitol than for those containing sucrose (2.1–8% and 8.5–16.6%, respectively; *p* < 0.05), indicating better stability of the structural properties of stored cookies with xylitol than of those containing sucrose.

The range of a_w_ in this study (0.36–0.46) was far below 0.6, the minimum a_w_ for the growth of microorganisms, indicating the high stability of stored cookies [42]. The microbiological quality of cookies, mold, and yeast, CFU/g (Colony Forming Units) was <10 for both sweeteners, independent of their amount in cookies or of the time of storage.

### 3.2. Profile of Volatile Compounds of Cookies

Application of the SPME/GC/MS method enabled the determination and identification of 46 volatile compounds in total in cookies containing different amounts of sucrose or xylitol, mostly aldehydes (11 compounds), hydrocarbons (10 compounds), and ketones (9 compounds). The multiway analysis of variance showed significant (*p* < 0.05) differences in relative amounts of many individual compounds and their total relative amounts in the type of sweetener and storage categories (Table 3). The content of sweetener did not significantly (*p* < 0.05) affect the relative amount of volatiles, thus this effect was not included in Table 3.

Regarding total relative amounts of volatiles from various categories, ketones aldehydes, hydrocarbons, and acids dominated the volatile profile of fresh cookies, while the cookies stored for 12 months were distinguished by high relative amounts of hydrocarbons, aldehydes, and acids, irrespective of the sweetener (Table 3).

Volatile compounds were categorized as follows: those detected only in fresh cookies (16 compounds), those detected in fresh cookies as well as in stored cookies (26 compounds), and those detected only in stored cookies (4 compounds). Detailed information is contained in Table 3. Many such compounds were identified in cookies or cakes containing diverse sweeteners (sucrose, glucose, or other sucrose replacers) [22,23,26,27].

The most abundant volatile detected only in fresh cookies was acetic acid, and its amounts were significantly (*p* < 0.05) higher in cookies containing sucrose compared with xylitol. Acetic acid is the product of the degradation of sugars via caramelization and Maillard reactions [27]. Cincotta et al. [22] reported acetic acid as a typical volatile compound in cookies containing sucrose, fructose, or glucose. Acetic acid may contribute to the pungent, sour, and vinegar odour notes of cookies [27,34]. In cookies containing xylitol, acetic acid was the likely substrate for the formation of butyl acetate ester, while butyl 3-methylbutyrate ester may be derived from butanoic (butyric) acid, a characteristic odorant for butter [44]. These compounds were not detected in fresh cookies containing sugar.

Among the Maillard reaction markers, pyrazines were the most numerous in fresh cookies. Pyrazines are formed in the early stages of the Maillard reaction as a result of α-dicarbonyl and amino acid reactions [23]. The presence of these compounds in cookies and cakes is favourable because of their contribution to imparting nutty and roasty aromas [23]. Three pyrazines, methylpyrazine, 2,5-dimethylpyrazine, and 2,3,5-trimethylpyrazine, detected in cookies containing sucrose or xylitol, were also identified by Garvey [23] in sucrose and reduced sucrose sponge cakes. In addition to those compounds, two other pyrazines were detected, and their presence was characteristic of the used sweetener: sucrose—3-ethyl-2,5-dimethylpyrazine and xylitol—2,6-dimethylpyrazine. Cepeda-Vázquez et al. [32] reported that key aroma compounds (Strecker aldehydes, pyrazines, and 2-pentylfuran) were positively related to the content of whole eggs, probably due to a higher contribution of free amino acids in egg yolk.

Furan compounds are the product of sugar dehydration/sugar fragmentation during the Maillard reaction or caramelization through direct decomposition of sugar moieties [23,45]. Furan has been classified as ‘possibly carcinogenic to humans’ (Group 2B) by the International Agency for Research on Cancer [46]. The furan and furan-derivative compounds are the product of heat processes (from 150 to 200 °C), especially during baking and roasting [22]. In contrast to other studies, only one furan compound (2-furancarboxaldehyde) was identified in fresh cookies containing only sucrose or xylitol.

More furans (6–9 compounds) were detected in sucrose and sucrose-reduced sponge cakes and biscuits containing different sugars—sucrose, fructose, or glucose [22,23]. The typical furanoic compound furfural was not detected. The differences between the presented results and those of other authors may have been due to the formulas used. Garvey et al. [23] and Cincotta et al. [22] used baking powder or ammonium bicarbonate, which increased pH. These ingredients were not used in this study; thus, the pH of the cookies ranged from 5.79 to 5.95 (Table 2). It was reported that pH values < 7 favour the formation of furan derivatives [34]. Moreover, at pH = 4.0 and 6.0, the addition of fructose was more effective for furan generation than the addition of glucose, sucrose, or sorbitol [47]. The reducing sugar glucose plays a crucial role in increasing furfural and furan formation in baked goods [32,44]. Nevertheless, thermal degradation of certain amino acids (serine, alanine, aspartic acid, threonine, and cysteine) and thermal oxidation of ascorbic acid and PUFAs may also contribute to furan generation [44]. Furan generation in sponge cake seems to proceed through common pathways to furfural (i.e., caramelization and Maillard reaction); their generation is enhanced in the presence of reducing sugars and hindered in the absence of egg yolk (probably due to differences in Maillard reaction progression, given the restriction on free amino acids [32].

Thus, it is possible that the use of sucrose or xylitol in formulas containing egg yolks at a pH of approximately 6.5 may be unfavourable for the production of furans in cookies. In addition, polyunsaturated fatty acids (PUFAs) and phospholipids present in egg yolk may exert a protective effect by forming antioxidant species when reacting with carbonyl compounds, hindering lipid oxidation [32].

**Table 3 foods-12-04270-t003:** Volatile compounds in cookies containing xylitol or sucrose (11, 17 & 23%)—fresh and stored for 12 months (means ± SD).

Volatile Compound (%)	Cc	LRI	CAS No	Odour Description	Cookies with Sucrose						Cookies with Xylitol						Xylitol vs. Sucrose	Storage Time, Sucrose	Storage Time, Xylitol
					Fresh			Stored for 12 Months			Fresh			Stored for 12 Months					
					11%	17%	23%	11%	17%	23%	11%	17%	23%	11%	17%	23%			
Cyclobutanol	Alc	-	2919-23-5	---	2.14 ± 0.16	1.84 ± 0.12	1.43 ± 0.09	n.d.	n.d.	n.d.	n.d.	n.d.	n.d.	n.d.	n.d.	n.d.	*	*	
Butanal	Ald	-	123-72-8	Pungent, cocoa, musty, malty, bready [48]	n.d.	n.d.	n.d.	n.d.	n.d.	n.d.	1.49 ± 0.06	1.53 ± 0.07	1.56 ± 0.10	n.d.	n.d.	n.d.	*		*
Phenylacetaldehyde	Ald	1043	122-78-1	Fatty, fruity, cake crust, bready [23]	0.24 ± 0.04	0.38 ± 0.01	0.40 ± 0.03	n.d.	n.d.	n.d.	0.34 ± 0.01	0.46 ± 0.07	0.47 ± 0.05	n.d.	n.d.	n.d.		*	*
4-Methyl-4-Hydroxy-2-Pentanone	K	836	123-42-2	---	1.81 ± 0.24	1.34 ± 0.13	3.37 ± 0.23	n.d.	n.d.	n.d.	1.55 ± 0.17	0.71 ± 0.09	1.46 ± 0.29	n.d.	n.d.	n.d.	*	*	*
5-Methyl-2-Hexanone	K	896	110-12-3	---	0.47 ± 0.06	0.43 ± 0.04	0.33 ± 0.07	n.d.	n.d.	n.d.	0.31 ± 0.04	0.31 ± 0.00	0.43 ± 0.06	n.d.	n.d.	n.d.		*	*
1-Phenylethanone (Acetophenone)	K	1065	98-86-2	Sweet, cake crust, burnt [23]	0.26 ± 0.12	0.42 ± 0.04	0.40 ± 0.05	n.d.	n.d.	n.d.	0.42 ± 0.01	0.36 ± 0.04	0.42 ± 0.00	n.d.	n.d.	n.d.		*	*
Acetic acid	Ac	-	64-19-7	Sharp, pungent, sour, vinegar [27]	7.93 ± 0.36	8.38 ± 0.69	11.33 ± 0.48	n.d.	n.d.	n.d.	6.97 ± 0.58	8.30 ± 0.16	7.74 ± 0.27	n.d.	n.d.	n.d.	*	*	*
Butyl acetate	E	813,5	123-86-4	Ethereal, solvent, fruity, banana [48]	n.d.	n.d.	n.d.	n.d.	n.d.	n.d.	0.19 ± 0.11	0.11 ± 0.04	0.28 ± 0.08	n.d.	n.d.	n.d.	*		*
Butyl 3-methylbutyrate	E	1050	109-19-3	---	n.d.	n.d.	n.d.	n.d.	n.d.	n.d.	0.22 ± 0.11	0.55 ± 0.08	0.62 ± 0.14	n.d.	n.d.	n.d.	*		*
Methylpyrazine	P	818	109-08-0	Nutty, cocoa, roasted, chocolate, peanut [48]	0.20 ± 0.06	0.29 ± 0.03	0.39 ± 0.10	n.d.	n.d.	n.d.	0.17 ± 0.10	0.54 ± 0.13	0.94 ± 0.10	n.d.	n.d.	n.d.		*	*
2,5-Dimethylpyrazine	P	910	123-32-0	cake crust, nutty, bready [23]	0.95 ± 0.31	1.67 ± 0.22	1.54 ± 0.01	n.d.	n.d.	n.d.	1.13 ± 0.19	1.11 ± 0.06	0.66 ± 0.03	n.d.	n.d.	n.d.		*	*
2,6-Dimethylpyrazine	P	915	108-50-9	Cocoa, coffee, green, roast beef, roasted nut, roasted [32]	n.d.	n.d.	n.d.	n.d.	n.d.	n.d.	0.34 ± 0.08	0.69 ± 0.01	0.82 ± 0.13	n.d.	n.d.	n.d.	*		*
2,3,5-Trimethylpyrazine	P	1000	14667-55-1	Cocoa, earthy, must, potato, roast [32]	0.65 ± 0.08	0.94 ± 0.06	1.10 ± 0.14	n.d.	n.d.	n.d.	0.41 ± 0.24	0.60 ± 0.01	0.58 ± 0.01	n.d.	n.d.	n.d.		*	*
3-Ethyl-2,5-Dimethylpyrazine	P	1069	13360-65-1	Potato, cocoa, roasted, nutty [48]	0.13 ± 0.04	0.27 ± 0.01	0.29 ± 0.01	n.d.	n.d.	n.d.	n.d.	n.d.	n.d.	n.d.	n.d.	n.d.	*	*	
1,4-Dichlorobenzene	Hc	1006	106-46-7	---	0.62 ± 0.05	0.62 ± 0.02	0.62 ± 0.04	n.d.	n.d.	n.d.	0.63 ± 0.08	0.60 ± 0.01	1.50 ± 0.01	n.d.	n.d.	n.d.		*	*
2-furancarboxy aldehyde	Fc	833	98-01-1	Spicy, bready [23]	0.12 ± 0.06	0.29 ± 0.01	0.37 ± 0.04	n.d.	n.d.	n.d.	0.04 ± 0.01	0.19 ± 0.04	0.20 ± 0.07	n.d.	n.d.	n.d.		*	*
Ethanol	Alc	-	64-17-5	Sweet, alcoholic [49]	2.88 ± 0.19	2.50 ± 0.28	2.03 ± 0.05	1.39 ± 0.06	1.40 ± 0.20	1.74 ± 0.32	2.42 ± 0.19	2.59 ± 0.13	1.38 ± 0.06	3.20 ± 0.35	2.29 ± 0.16	2.20 ± 0.02		*	
3-Methylbutanal	Ald	643	590-86-3	Musty, cocoa, coffee, nutty [27]	4.51 ± 0.89	4.29 ± 0.28	5.85 ± 0.35	1.64 ± 0.31	1.90 ± 0.01	1.69 ± 0.13	4.42 ± 0.64	5.42 ± 0.23	7.87 ± 0.31	2.03 ± 0.02	2.84 ± 0.45	2.40 ± 0.45	*	*	*
2-Methylbutanal	Ald	653,5	96-17-3	Ethereal, aldehydic, chocolate, peach, fatty [28]	4.57 ± 0.81	5.09 ± 0.42	6.00 ± 0.21	3.00 ± 0.44	3.73 ± 0.05	4.43 ± 0.41	5.18 ± 0.33	6.14 ± 0.41	7.87 ± 0.12	3.12 ± 0.07	4.21 ± 0.58	5.02 ± 0.17	*	*	*
Pentanal	Ald	696	110-62-3	Fermented, bready, fruity, nutty, berry [48]	4.23 ± 0.32	3.73 ± 0.24	3.41 ± 0.09	3.10 ± 0.32	3.29 ± 0.17	3.85 ± 0.41	4.54 ± 0.24	4.21 ± 0.11	3.71 ± 0.03	6.02 ± 0.48	6.03 ± 0.61	4.54 ± 0.05	*	*	*
Hexanal	Ald	798	66-25-1	Green, fatty, leafy, fruity, aldehydic, sweaty, grass [27]	5.25 ± 0.36	3.89 ± 0.45	3.89 ± 0.07	0.99 ± 0.09	0.87 ± 0.14	1.44 ± 0.25	5.64 ± 0.15	5.75 ± 0.52	4.00 ± 0.13	1.53 ± 0.03	1.79 ± 0.12	2.16 ± 0.66	*	*	*
Heptanal	Ald	901	111-71-7	Fresh, green, sweet, herbal, wine-lee, ozone [27]	2.02 ± 0.14	1.47 ± 0.02	1.24 ± 0.05	0.34 ± 0.05	0.36 ± 0.01	0.60 ± 0.06	1.69 ± 0.21	1.66 ± 0.01	1.33 ± 0.04	0.58 ± 0.05	0.53 ± 0.02	0.54 ± 0.04	*	*	*
Benzaldehyde	Ald	960	100-52-7	Strong, sharp, sweet, bitter, almond, cherry [27]	0.58 ± 0.11	1.30 ± 0.13	0.42 ± 0.02	0.13 ± 0.06	0.17 ± 0.03	0.28 ± 0.04	0.97 ± 0.15	0.93 ± 0.01	0.89 ± 0.08	0.21 ± 0.05	0.26 ± 0.07	0.28 ± 0.02	*	*	*
Octanal	Ald	1003	124-13-0	Aldehydic, waxy, citrus, orange, green, peel [27]	0.93 ± 0.01	0.94 ± 0.01	1.00 ± 0.06	1.12 ± 0.37	1.06 ± 0.30	0.95 ± 0.01	0.83 ± 0.18	0.89 ± 0.04	0.65 ± 0.00	1.24 ± 0.02	0.91 ± 0.07	0.89 ± 0.02		*	*
Nonanal	Ald	1104	124-19-6	Bready, cake crust [23]	5.35 ± 0.46	4.09 ± 0.06	3.73 ± 0.06	5.92 ± 0.05	7.92 ± 0.27	4.61 ± 0.17	5.42 ± 0.21	4.46 ± 0.19	2.39 ± 0.03	3.22 ± 0.21	3.29 ± 0.15	3.39 ± 0.32	*	*	*
Decanal	Ald	1206	112-31-2	Sweet, waxy, orange peel, citrus, floral [27]	0.32 ± 0.01	0.24 ± 0.00	0.16 ± 0.23	0.28 ± 0.05	0.29 ± 0.01	0.40 ± 0.06	n.d.	n.d.	n.d.	0.31 ± 0.04	0.35 ± 0.01	0.45 ± 0.02	*	*	*
Acetone (2-Propanon)	K	-	67-64-1	---	5.11 ± 0.47	4.58 ± 0.24	3.76 ± 0.46	n.d.	n.d.	n.d.	9.23 ± 0.71	8.51 ± 0.59	8.37 ± 0.37	5.18 ± 0.70	6.57 ± 0.66	5.68 ± 0.40	*	*	*
2-Pentanone	K	681,5	107-87-9	---	2.77 ± 0.14	2.43 ± 0.25	2.56 ± 0.11	n.d.	n.d.	n.d.	2.24 ± 0.12	2.24 ± 0.08	2.72 ± 0.05	0.59 ± 0.26	0.82 ± 0.02	0.91 ± 0.28	*	*	*
3-Hydroxy-2-Butanone (Acetoin)	K	706	513-86-0	Buttery, creamy, dairy, milky, fatty, sweet yogurt [50]	3.11 ± 0.33	1.97 ± 0.16	2.02 ± 0.51	0.87 ± 0.13	0.87 ± 0.14	1.17 ± 0.27	2.49 ± 0.57	3.60 ± 1.00	2.07 ± 0.23	1.50 ± 0.49	2.58 ± 0.14	2.15 ± 0.72	*	*	*
2-Heptanone	K	890	110-43-0	Fruity, sweet [50]	15.69 ± 0.71	16.35 ± 0.68	13.47 ± 0.55	0.11 ± 0.02	0.17 ± 0.01	0.22 ± 0.04	13.28 ± 0.07	13.74 ± 0.35	17.86 ± 0.36	0.33 ± 0.04	0.26 ± 0.02	0.27 ± 0.01		*	*
2-Nonanone	K	1093	821-55-6	Fruity, hot milk [50]	7.36 ± 0.62	8.17 ± 0.46	8.55 ± 0.06	0.79 ± 0.02	1.09 ± 0.05	1.52 ± 0.03	7.44 ± 0.32	7.81 ± 0.32	6.22 ± 0.23	0.28 ± 0.03	0.29 ± 0.06	0.34 ± 0.03	*	*	*
2-Undecanone	K	1294	112-12-9	Fruity, waxy [48]	1.30 ± 0.03	1.54 ± 0.08	1.88 ± 0.17	0.88 ± 0.11	1.12 ± 0.13	1.45 ± 0.31	1.42 ± 0.08	1.76 ± 0.13	1.29 ± 0.17	0.63 ± 0.06	0.83 ± 0.03	0.84 ± 0.11		*	*
Butanoic acid	Ac	788	107-92-6	Sharp, sour, acetic, cheesy [48]	1.98 ± 0.39	2.50 ± 1.04	1.74 ± 0.36	9.16 ± 0.56	5.11 ± 0.22	7.93 ± 0.98	0.72 ± 0.28	0.53 ± 0.12	0.34 ± 0.09	8.60 ± 0.27	4.94 ± 0.10	3.24 ± 0.27	*	*	*
Hexanoic acid	Ac	994	142-62-1	Mild, sour, fatty, sweet, cheesy [27]	0.91 ± 0.17	3.20 ± 1.10	5.12 ± 0.77	6.15 ± 0.91	5.46 ± 0.73	4.48 ± 0.25	1.18 ± 0.05	1.01 ± 0.25	0.94 ± 0.05	5.92 ± 0.75	2.70 ± 0.09	2.75 ± 1.34	*	*	*
α-(+)-Pinene	T	939	80-56-8	Fresh, camphoreous, sweet, pine, earthy, woody [48]	0.35 ± 0.06	0.44 ± 0.11	0.51 ± 0.24	0.04 ± 0.00	0.05 ± 0.00	0.05 ± 0.01	0.38 ± 0.12	0.32 ± 0.01	0.32 ± 0.06	0.11 ± 0.02	0.08 ± 0.01	0.08 ± 0.03	*	*	*
D-Limonene	T	1026	5989-27-5	Citrus, orange, fresh, sweet, [48]	2.84 ± 0.30	3.06 ± 0.18	2.54 ± 0.10	0.28 ± 0.05	0.37 ± 0.16	0.42 ± 0.07	3.43 ± 0.13	2.85 ± 0.06	2.21 ± 0.05	0.33 ± 0.05	0.26 ± 0.03	0.45 ± 0.02	*	*	*
Cyclopentane	Hc	-	287-92-3	---	5.30 ± 0.43	4.07 ± 0.28	2.57 ± 0.71	1.53 ± 0.20	1.69 ± 0.31	1.71 ± 0.23	4.93 ± 0.95	2.52 ± 0.11	4.05 ± 0.14	2.37 ± 0.38	2.42 ± 0.14	2.17 ± 0.15		*	*
2,2,4,6,6-Pentamethylheptane	Hc	985,5	13475-82-6	---	4.37 ± 0.23	4.83 ± 0.28	3.85 ± 0.11	0.13 ± 0.00	0.19 ± 0.01	0.24 ± 0.01	3.88 ± 0.18	3.76 ± 0.03	3.04 ± 0.01	0.11 ± 0.01	0.17 ± 0.04	0.22 ± 0.03	*	*	*
Dodecane	Hc	1200	112-40-3	Alkane [27]	0.19 ± 0.01	0.25 ± 0.03	0.33 ± 0.02	0.31 ± 0.05	0.37 ± 0.01	0.49 ± 0.08	0.23 ± 0.04	0.21 ± 0.04	0.15 ± 0.03	0.33 ± 0.03	0.32 ± 0.01	0.49 ± 0.07	*	*	*
Toluene	Hc	755	108-88-3	---	1.12 ± 0.01	0.87 ± 0.02	0.60 ± 0.01	0.34 ± 0.13	0.56 ± 0.02	0.57 ± 0.01	1.97 ± 0.02	1.45 ± 0.10	1.37 ± 0.02	0.59 ± 0.11	0.49 ± 0.08	0.54 ± 0.03	*	*	*
1,4-Dimethylbenzene (1,4-Xylene; P-Xylene)	Hc	864	106-42-3	---	0.63 ± 0.05	0.66 ± 0.14	0.65 ± 0.12	0.13 ± 0.02	0.13 ± 0.00	0.21 ± 0.04	0.77 ± 0.13	0.71 ± 0.01	0.51 ± 0.11	0.25 ± 0.01	0.16 ± 0.00	0.17 ± 0.02	*	*	*
Styrene (Ethenylbenzene)	Hc	887	100-42-5	---	0.81 ± 0.06	0.78 ± 0.05	0.56 ± 0.05	0.14 ± 0.01	0.12 ± 0.01	0.14 ± 0.03	1.20 ± 0.01	0.98 ± 0.06	0.77 ± 0.03	0.10 ± 0.00	0.09 ± 0.03	0.35 ± 0.03	*	*	*
Octanoic acid	Ac	1182	124-07-2	Fatty, acid, sour [27]	n.d.	n.d.	n.d.	2.56 ± 0.30	0.48 ± 0.10	0.88 ± 0.15	n.d.	n.d.	n.d.	n.d.	n.d.	n.d.	*	*	
Pentane	Hc	-	109-66-0	---	n.d.	n.d.	n.d.	7.25 ± 0.52	2.96 ± 0.62	4.70 ± 0.21	n.d.	n.d.	n.d.	n.d.	n.d.	n.d.	*	*	
Hexane	Hc	598	110-54-3	---	n.d.	n.d.	n.d.	51.20 ± 0.47	58.05 ± 0.00	53.58 ± 0.62	n.d.	n.d.	n.d.	50.93 ± 0.37	54.12 ± 1.66	57.22 ± 1.44		*	*
Tetradecane	Hc	1399	629-59-4	Mild, waxy [27]	n.d.	n.d.	n.d.	0.22 ± 0.02	0.23 ± 0.00	0.32 ± 0.08	n.d.	n.d.	n.d.	0.37 ± 0.01	0.36 ± 0.03	0.30 ± 0.04	*	*	*
Total alcohols					5.02 ± 0.03	4.33 ± 0.47	3.46 ± 0.04	1.39 ± 0.06	1.40 ± 0.20	1.74 ± 0.32	2.42 ± 0.19	2.59 ± 0.13	1.38 ± 0.06	3.20 ± 0.35	2.29 ± 0.16	2.20 ± 0.02	*	*	*
Total aldehydes					27.99 ± 1.60	25.40 ± 1.23	26.08 ± 0.25	16.52 ± 1.00	19.58 ± 0.52	18.23 ± 0.45	30.50 ± 0.65	31.43 ± 0.44	30.73 ± 0.03	18.27 ± 0.22	20.23 ± 0.93	19.65 ± 1.34	*	*	*
Total ketones					37.88 ± 1.65	37.21 ± 1.35	36.34 ± 0.52	2.65 ± 0.02	3.26 ± 0.05	4.36 ± 0.64	38.36 ± 1.34	39.01 ± 0.35	40.85 ± 0.26	8.52 ± 0.10	11.37 ± 0.54	10.19 ± 1.55	*	*	*
Total acids					10.82 ± 0.92	14.07 ± 0.75	18.19 ± 0.89	17.87 ± 1.16	11.05 ± 0.40	13.28 ± 1.07	8.86 ± 0.25	9.83 ± 0.28	9.03 ± 0.30	14.52 ± 0.48	7.64 ± 0.01	5.99 ± 1.60	*	*	*
Total esters					n.d.	n.d.	n.d.	n.d.	n.d.	n.d.	0.40 ± 0.21	0.66 ± 0.13	0.90 ± 0.22	n.d.	n.d.	n.d.	*		*
Total pyrazines					1.93 ± 0.61	3.16 ± 0.59	3.32 ± 0.25	n.d.	n.d.	n.d.	2.04 ± 0.61	2.93 ± 0.21	3.00 ± 0.26	n.d.	n.d.	n.d.	*	*	*
Total terpenes					3.19 ± 0.36	3.50 ± 0.29	3.05 ± 0.35	0.32 ± 0.05	0.42 ± 0.16	0.47 ± 0.06	3.81 ± 0.25	3.16 ± 0.07	2.53 ± 0.12	0.43 ± 0.07	0.34 ± 0.04	0.53 ± 0.01	*	*	*
Total hydrocarbons					13.04 ± 0.12	12.06 ± 0.21	9.18 ± 0.99	61.25 ± 0.17	64.3 ± 0.30	61.94 ± 0.90	13.59 ± 1.36	10.22 ± 0.21	11.38 ± 0.01	55.05 ± 0.08	58.13 ± 1.34	61.45 ± 1.25	*	*	*

Cc—chemical classes of volatile compounds; Alc—Alcohols; Ac—Carboxylic acids; Ald—Aldehydes; Hc—Hydrocarbons; E—Esters; K—Ketones; T—Terpenes; P—Pyrazines; Fc—Furan compounds; n.d.—not detected; * Significantly different (*p* < 0.05) depending on the sweetener (xylitol vs. sucrose) or storage time. LRI- calculated.

Regarding other volatiles detected only in fresh cookies (Table 3), phenylacetaldehyde derived from Strecker degradation from the amino acid phenylalanine may contribute to the sweet, rose, and honey aromas of sponge cakes [23,51]. The other volatile, butanal, arises from the lipid oxidation of linoleic acid [52].

Most of the detected aldehydes were previously identified in cookies and cakes; 3-methylbutanal and 2-methylbutanal probably result from Strecker degradation reactions of branched-chain amino acids—valine and leucine, respectively [22,23,27]. This family of precursors is present in egg yolk [27]. Although previous studies indicated that mono- and disaccharides are important substrates of 3-methylbutanal and 2-methylbutanal in cookies, substantial amounts were also assayed in xylitol-containing cookies [23,27] (Table 3). Among Strecker aldehydes, benzaldehyde was also detected; however, it had a much lower relative abundance than the two mentioned compounds and was probably derived from phenylalanine, which is also contained in eggs (Table 3) [23]. All three aldehydes are regarded as odour-active compounds and contribute to the sweet, toffee, butterscotch, and pineapple aromas of cookies and cakes [22,23]. The other six aliphatic aldehydes in the volatile profiles of cookies were probably derived from lipid oxidation processes [34]. Heptanal, octanal, nonanal, and decanal could be formed by autooxidation of oleic acid contained in butter—the main ingredient of cookies, while hexanal and pentanal are regarded as the products of oxidative degradation of linoleic acid [27,53]. All these aldehydes are among the volatiles most highly positively correlated with the perception of rancidity in oat cakes [53].

Volatile profiles of fresh and stored cookies were characterized by a substantial abundance of methyl ketones: six compounds were detected in fresh as well as in stored cookies, and three compounds were detected exclusively in fresh cookies (Table 3). Methyl ketones may be formed primarily by hydrolytic reactions under the action of heat and moisture [54]. These compounds may also be formed from unsaturated fatty acids and their unsaturated secondary degradation products resulting from lipid oxidation and can be considered tertiary products of lipid oxidation [55]. Irrespective of the sweetener used, fresh cookies contained the highest relative amounts of 2-heptanone: 15.2% and 14.95% on average in cookies containing sucrose and xylitol, respectively. 2-Nonanone, 2-pentanone, and 3-hydroxy-2-butanone were also quantitatively important volatiles in fresh and stored cookies. Many ketones detected in cookies were odour-active in fresh or heated butter, contributing to creamy, blue cheese, and fruity notes [28]. These methyl ketones may be derived from short-chain and middle-chain fatty acids shorter than 14 C atoms by enzymatic hydrolysis and by β-oxidation of triacylglycerols and are typically found in milk fat [22,56]. The 3-hydroxy-2-butanone, giving a buttery aroma, is probably derived from the conversion of 2,3-butanedione (diacetyl) [57]. Acetone (2-propanone) was present in cookies containing either sweetener, and its relative amounts were twice as high in fresh xylitol-containing cookies compared to those with sucrose (8.7% and 4.5%, respectively).

Other ketones, 4-methyl-4-hydroxy-2-pentanone and 5-methyl-2-hexanone, detected only in fresh cookies, are probably tertiary degradation products of lipid oxidation [55]. Acetophenone was found in sucrose and sucrose-reduced sponge cakes and is regarded as the most odour active in sponge cakes, contributing to their sweet and cake crust aroma [23].

Hydrocarbons in fresh and stored cookies were represented by six compounds, including aromatic, linear, and cyclic compounds, and their derivatives (Table 3). The relative contents of cyclopentane and 2,2,4,6,6-pentamethylpentane were much higher than those of toluene, styrene, p-xylene, and dodecane. The presence of hydrocarbons in cookies and cakes probably resulted from lipid oxidation [23,27,34]. Thermal decomposition of yolk-containing carotenoids and phospholipids during baking may be the source of toluene, styrene, and p-xylene. However, hydrocarbons do not greatly affect the aroma of cookies because of their high odour thresholds.

The storage of cookies for 12 months led to changes in the relative contents of many volatile compounds and generated the formation of new compounds (Table 3, Figure 1). The organic acids butanoic (C4:0) and hexanoic (C6:0) generally showed a 2–5-fold increase in stored cookies compared with fresh cookies, irrespective of sweetener type. Such an increase is probably derived from the lipolysis of butter during storage. In sucrose-containing cookies, the lipolytic process liberated octanoic acid (C8:0), which was absent in cookies with xylitol, probably because of the antioxidative potential of the sweetener [58].

Storage of cookies decreased the relative contents of linear aldehydes (e.g., hexanal, heptanal), in both cookies containing xylitol or sucrose. As reported by Grebenteuch et al. [55], aldehydes such as hexanal are suitable as markers for the early phase of lipid oxidation, but in the advanced stage, they are subject to changes by degradation to form other compounds. The cookies stored with sucrose did not contain two ketones, 2-propanone (acetone) and 2-pentanone, while in xylitol cookies, their levels were quite meaningful, averaging approximately 5.80% and 2.30%, respectively (Table 4). Acetone is regarded as an indicator of lipid oxidation in stored margarine, yet trace amounts of acetone were detected in cookies containing sucrose [23,56].

In stored cookies, the relative levels of three Strecker aldehydes decreased, while other Maillard reaction products, e.g., pyrazines or acetic acid, were not detected. Such observations were reported for stored cookies containing sucrose and enriched with freeze-dried Japanese quince fruits [34].

The most important quantitative and qualitative changes in the volatile profiles of cookies during storage pertained to the hydrocarbon group (Table 3, Figure 1). Storage induced the generation of three new hydrocarbons not detected in fresh cookies: pentane, hexane, and tetradecane. Throughout the period of 0–9 months of storage, the total content of hydrocarbons was stable and averaged 10.2% in cookies with xylitol and 12.8% in cookies with sucrose (Figure 1). A substantial increase in the total content of hydrocarbons was found in xylitol and sucrose cookies stored for 12 months (58.2% and 60.35%, respectively). Such an increase in the total content of hydrocarbons was due to the formation of hexane, whose relative content exceeded 50%, irrespective of the type and content of the sweetener. These results are in accordance with the study of Antoniewska et al. [34], who stated that hydrocarbons may be regarded as process-related contaminants of cookies. The formation of hydrocarbons may have resulted from the decomposition of unstable hydroperoxides, whose degradation results from the formation of secondary products, e.g., aliphatic aldehydes and hydrocarbons. The obtained results suggested that the 9-month storage period of cookies, under the described conditions (no additives extending shelf life had been added), was the recommended period regarding the safety of cookies.

### 3.3. Descriptive Sensory Analysis

Cookies containing xylitol differed significantly (*p* < 0.05) from those containing sucrose with regard to the descriptive test attributes. The storage of cookies significantly affected the rating intensity of many sensory indices (Table 4).

When rating the intensity of aroma attributes, the buttery aroma was perceived as dominant in fresh cookies containing xylitol, and its intensity was similar to that in sucrose cookies (5.66–6.34 and 5.66–6.75, respectively; Table 4). The buttery aroma probably resulted from the presence of ketones, especially 3-hydroxy-2-butanone (acetoin; Table 3), as reported by Mallia et al. [28]. A relatively high content of 2-heptanone (approximately 15% on average), a characteristic odour factor of heated butter, may suggest its crucial role in the buttery aroma of cookies [28]. However, that compound was not found to be odour-active in all types of cakes [23]. The sensory panel scored decreased intensity of buttery aroma in cookies stored for 9 months, probably due to a high decrease in volatile compounds responsible for buttery aroma (Table 4, Figure 2) [34]. The scored intensity of buttery aroma decreased more in sucrose-containing cookies than in xylitol-containing cookies.

Irrespective of the type of sweetener, fresh cookies were highly rated with respect to the roasty aroma: 4.90–7.30 and 4.60–6.27 scores for cookies containing xylitol and sucrose, respectively (Table 4). This scoring probably resulted from the presence of many pyrazines detected by the GC/MS/SPME assay, which enabled the detection of the roasty aroma of the cookies (Table 3). Among the five detected pyrazines, one of them (3-ethyl-2,5-dimethylpyrazine) contained an ethyl group, which increased the perception of the roasty aroma of the cookies [29]. Another compound, nonanal, present in relatively high amounts, may have contributed to the roasty aroma. Garvey et al. [23] reported that high levels of nonanal in sponge cakes gave them “bready” and “cake crust” impressions.

A high decrease in the scoring intensity of roasty aroma was noted in stored cookies (3.05–4.13 and 2.85–3.82 for xylitol and sucrose cookies, respectively). As the odour thresholds of pyrazines are low, they induce a roasty aroma even at low levels.

The sensory panel also indicated sweet/caramelized as an important aroma attribute of xylitol-containing cookies, probably due to high levels of Strecker aldehydes (3-methylbutanal and 2-methylbutanal). The sweet/caramelized aroma may also be due to the presence of phenylacetaldehyde, 2-furacarboxaldehyde, and acetophenone, which are found in sucrose and sucrose-reduced sponge cakes [23]. Fresh cookies containing xylitol or sucrose differed significantly (*p* < 0.05) in the intensity of sweet/caramelized aroma, but an approximately 2-fold decrease resulted from storage (Table 4, Figure 2).

The nutty aroma may be regarded as the next important attribute likely induced by pyrazines, with intensities of 2.63–3.32 and 2.35–3.04 points for cookies containing xylitol or sucrose, respectively. Higher scores of nutty aroma are related to Maillard and caramelization reactions of sucrose [23].

The high content of Strecker aldehydes (3-methylbutanal and 2-methylbutanal) may contribute to the fruity aroma. The intensity of the fruity aroma in fresh cookies did not exceed 2.1 points, probably due to the dominance of other aroma attributes (buttery and caramel) (Table 4). In stored cookies, the fruity aroma was even less perceptible, with an intensity of 0.40–0.78 points.

The sour note in the aroma of fresh cookies was primarily due to the presence of acetic acid, whose levels were higher in sucrose cookies than in xylitol cookies (Table 3 and Table 4). Storage increased the scoring intensity of sour aroma. It was more noticeable in sucrose-containing cookies than in those with xylitol (Table 4, Figure 2). It may have resulted from a higher increase in pH values and relative contents of organic, butyric, and caproic acids in cookies with sucrose than in cookies containing xylitol (Table 2 and Table 3).

Generally, pungency was a minor attribute in the aroma profile of fresh cookies, and storage led to an approximately four-fold increase in scoring the sucrose-containing cookies, much more than in xylitol-containing cookies. An increase in the relative contents of organic acids (butanoic and hexanoic) and the formation of octanoic acid in stored sucrose cookies could have resulted in increased ratings of their pungent aroma (Table 3 and Table 4).

The type of sweetener significantly (*p* < 0.05) affected the colour intensity of cookies, from yellow (xylitol) to brown (sucrose), probably because sucrose accelerates the Maillard and caramelization reactions during baking [23,29,59]. A brighter colour of cookies in which sucrose was partly replaced with polyol (erythritol) compared with that of sucrose-containing cookies was also reported by Lin et al. [18]. As anticipated, we observed no significant (*p* < 0.05) changes in scoring colour intensity for both types of cookies during storage (Table 4). Similar results were obtained for cookies stored with sucrose supplemented with tannin-encapsulated emulsion [43]. 

The texture attributes significantly (*p* < 0.05) depended on the type of sweetener (Table 4). Fresh xylitol cookies were approximately two-fold less hard and fracturable than sucrose cookies (Table 4), as reported by Winkelhausen et al. [21], who showed that sucrose cookies were harder, drier, and crunchier than xylitol cookies. Similar results were also obtained by Zoulias et al. [19], who found xylitol cookies to be three-fold less hard than sucrose cookies. Lin et al. [18] revealed that sucrose acts as a hardening agent by crystallizing as the cookie cools, making the product crisp. The softening effect of biscuits when sugars were substituted with sugar alcohols was reported by Olinger and Velasco [60]. When erythritol was combined with fats, e.g., butter, the cooling effect made the texture waxy [18]. It should also be noted that a higher content of sweetener resulted in panellists perceiving a higher intensity of hardness of fresh cookies, irrespective of the sweetener (Table 4).

Adhesiveness and moisture were rated differently from hardness and fracturability. Higher ratings were indicated for fresh xylitol cookies than for sucrose cookies, probably because xylitol has five hydroxyl groups (OH) that may interact with water by hydrogen bonds, hindering the evaporation process during baking, which may be related to the higher adhesiveness and moisture of cookies [61]. Although sucrose is a humectant binding moisture, xylitol was found to be a stronger humectant than sucrose in cookies [62]. This view was supported by water activity assays of fresh xylitol cookies revealing higher water activity than sucrose cookies (on average a_w_ = 0.366 and 0.346, respectively) (Table 2). Xylitol has greater potential as a humectant than sorbitol because of its much lower glass transition temperature (Tg), making it more effective at softening protein and starch matrices [61].

Storage significantly (*p* < 0.05) decreased the hardness of sucrose cookies (Figure 3). Similarly, stored sucrose cookies, but not xylitol cookies, were scored as less fracturable than fresh cookies (Table 4). These differences may be related to the values of a_w_, and a higher increase was found in cookies containing sucrose than in cookies with xylitol (Table 2).

Irrespective of the type of sweetener, stored cookies had higher ratings of moisture and adhesiveness than fresh cookies (Figure 3, Table 4), associated with increased values of a_w_ in stored cookies (a_w_ = 0.389, on average; Table 2). The texture characteristics of cookies were also strongly dependent on the properties of packaging materials and their permeability; moreover, textural parameters may be affected by changes in moisture content and can be mainly attributed to the phenomena of migration of moisture and its redistribution within the sample [42].

The fundamental attribute of cookies, sweet taste, significantly (*p* < 0.05) depended on the sweetener, with xylitol cookies being rated as less sweet than sucrose cookies (Table 4). Similar results were reported by Zoulias et al. [19]. Most likely, xylitol has a relative sweetness (RS) similar to that of sucrose but appears quickly and has a shorter persistence [9,63]. Reduced sweetness of cookies with sucrose replaced in the range of 50–100% by polyol (erythritol) was reported by Lin et al. [18]. In contrast to our results, Winkelhausen et al. [21] reported that the sweetness of xylitol cookies was rated the same as that of sucrose cookies. The inclusion of xylitol in the cookie recipe also resulted in decreased scoring of other taste attributes, buttery and roasty, compared with sucrose cookies, probably as an effect of menthol aftertaste (Table 4, Figure 4).

Generally, the storage of cookies significantly (*p* < 0.05) affected the rating intensities of many taste attributes (Table 4). Stored cookies were rated lower regarding buttery and roasty tastes compared with fresh cookies, irrespective of the type of sweetener, and the perception of sweet taste in stored cookies remained at the same level as that in fresh cookies.

Since xylitol may impart a cooling effect in food products, the panellists decided to include the cooling effect as an aftertaste on the list of estimated attributes (during the session devoted to the generation of sensory attributes). The panellists were able to discriminate cookies containing different sweeteners. The cooling effect was perceived in xylitol cookies (2.80–3.90 scores) and significantly decreased during storage (1.85–3.10 scores) while scoring sucrose cookies was negligible (Table 4, Figure 4).

To better understand the effects of sugar type and content and storage on the assessment of sensory attributes, PCA was applied, its effects significantly varying between samples (see Table 4). Regarding the scores plot, there is a clear separation between fresh cookies and those stored for 12 months, containing different amounts of xylitol or sucrose (Figure 5). The variables that were mainly correlated on the negative side of PC1 and PC2 were adhesiveness, moisture, and cooling effect, which characterised stored xylitol cookies. Aroma attributes: fatty/rancid, pungent, and sour, correlated on the positive PC2 and negative PC1, were associated with stored sucrose cookies. Sensory attributes of fresh cookies were located on the positive sides PC2 and PC2. Fresh cookies with sucrose were associated with sweet, buttery, and roasted tastes, hardness and fracturability, and colour. The sensory profile of fresh cookies with xylitol was represented mainly by positive aromas (buttery, nutty, roasted/baked, and sweet/caramelized).

As shown in Figure 5, on the cluster analysis dendrogram, two different clusters were seen depending on the type of sweetener and were also separated according to storage (fresh and storage), resulting in four groups to be distinguished (C1, C2, C3, and C4).

In summary, the use of xylitol in the recipe had a positive effect on maintaining the sensory quality of cookies during storage, e.g., enabled preservation of the buttery aroma to a greater extent than in the case of sucrose cookies; it also prevented the formation of sour and pungent notes during storage, which were perceived in cookies with sucrose. Stored cookies with xylitol retained their hardness and fracturability to a greater extent than the cookies with sucrose.

## 4. Conclusions

Analyses conducted using the GC/MS/SPME technique showed similarities in the volatile compound profiles of xylitol- or sucrose-containing cookies. Many compounds assayed in cookies containing xylitol were recognized as markers of Maillard reactions (Strecker aldehydes, and pyrazines), typical for changes in which sugars are the substrate. The similarities applied also to the levels of compounds unfavourably contributing to the volatile profiles of cookies (aldehydes, hydrocarbons, and organic acids).The study showed that xylitol had better properties than sucrose as an ingredient in cookies, positively affecting their shelf life. Unlike sucrose, the use of xylitol in the recipe of the “clean label” affected the stability of pH and water activity.The buttery aroma was perceived as dominant in fresh cookies containing xylitol, its intensity being similar to that in sucrose cookies (5.66–6.34 and 5.66–6.75 points, respectively. The use of xylitol enabled the maintenance of some sensory attributes, such as buttery aroma and texture characteristics, and limited the formation of unfavourable aromas (such as fatty/rancid) during 12 months of storage.Until 9 months of storage, the volatile profiles of cookies contained low amounts of compounds regarded as contaminants. After the 9th month of storage, the hydrocarbons greatly increased (on average, 58.2% and 60.35% in xylitol and sucrose cookies, respectively, stored for 12 months). The results indicated that under the conditions assumed in this study, 9 months is the maximum recommended period of storage for cookies due to safety, without using any additives to extend shelf life.

## Figures and Tables

**Figure 1 foods-12-04270-f001:**
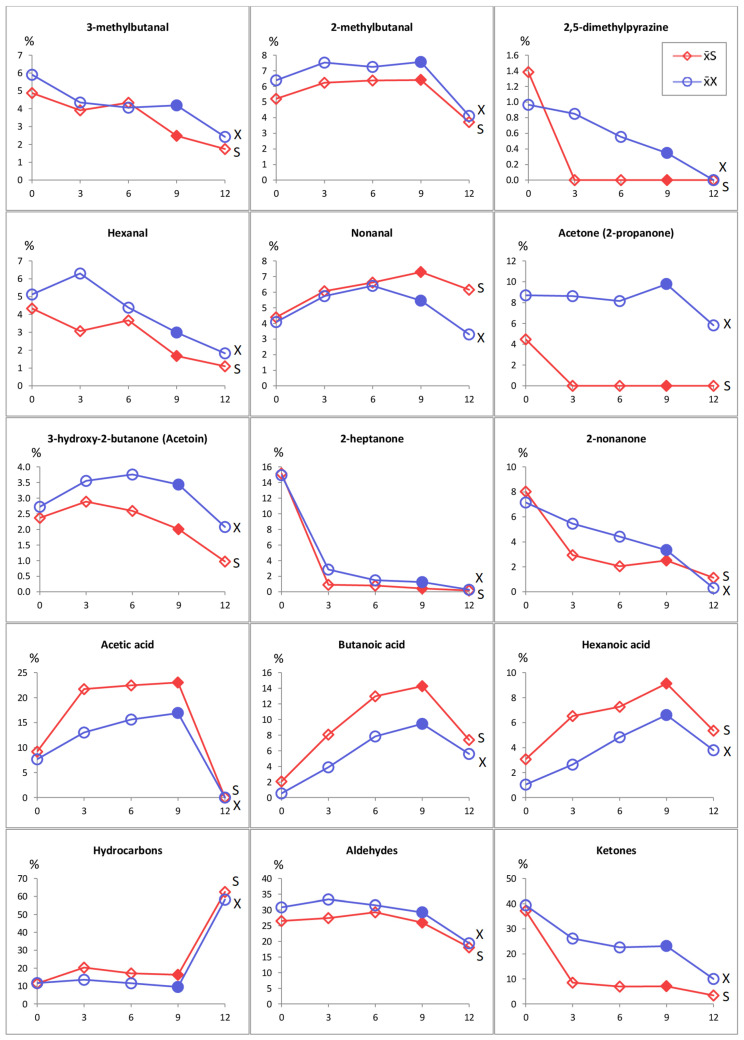
Changes in relative levels of volatile compounds in cookies containing xylitol—
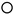

or sucrose—
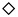
 during 12 months of storage. The changes in the relative abundance of volatiles are significant (*p* < 0.05) depending on the storage period.

**Figure 2 foods-12-04270-f002:**
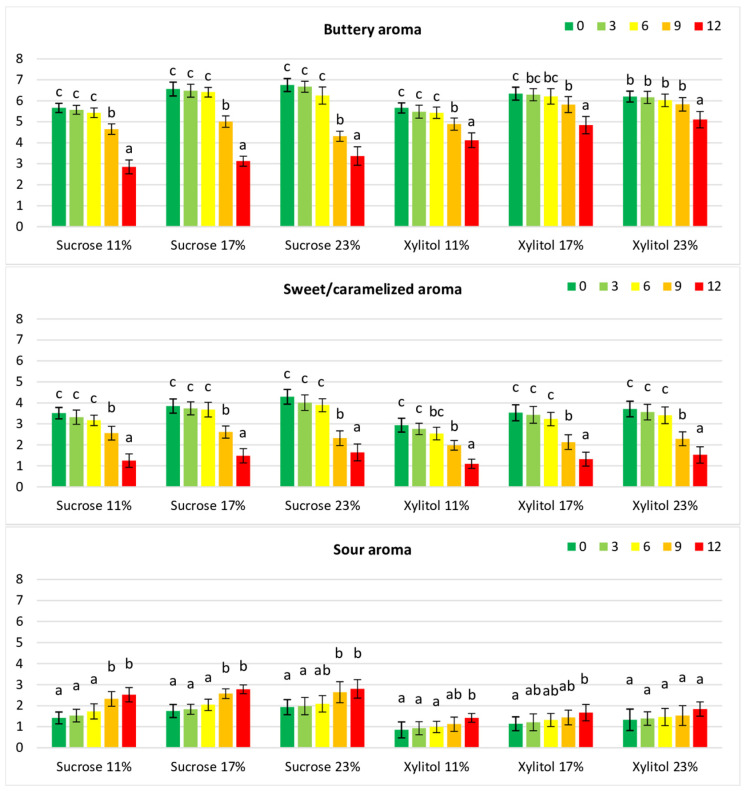
**Sensory** intensity of selected aroma attributes of cookies containing xylitol or sucrose during storage. Values for the cookie type with different superscripts (a, b, c) differ significantly (*p* < 0.05).

**Figure 3 foods-12-04270-f003:**
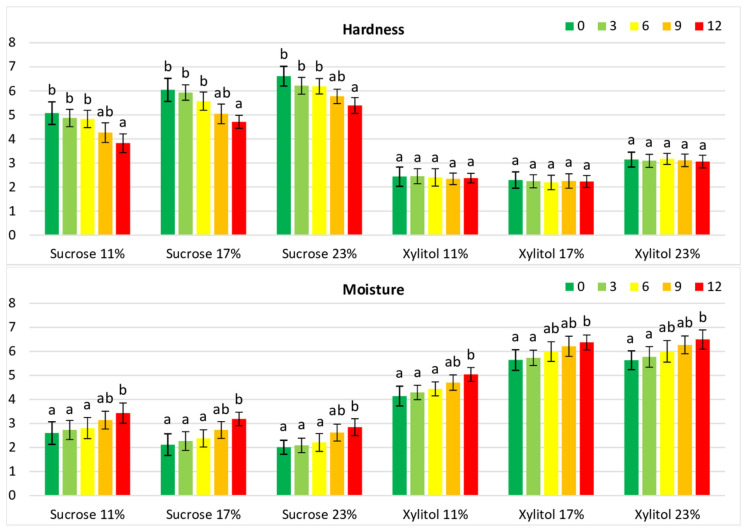
Sensory intensity of hardness and moisture of cookies containing xylitol or sucrose during storage. Values for the cookie type with different superscripts (a, b) differ significantly (*p* < 0.05).

**Figure 4 foods-12-04270-f004:**
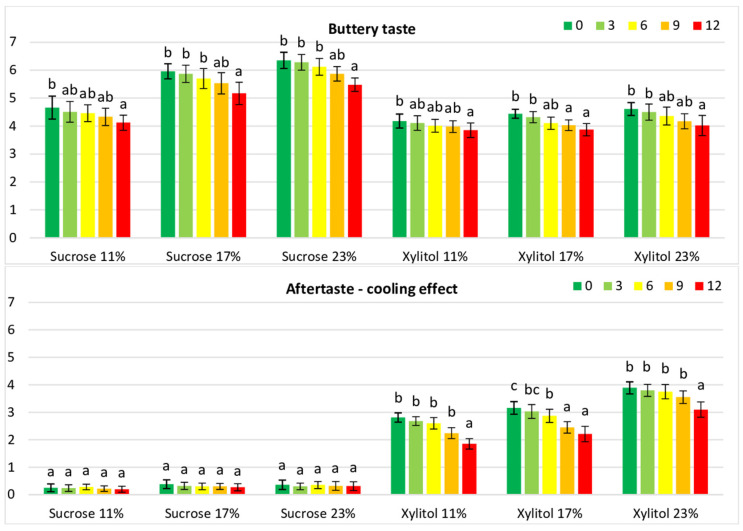
Sensory intensity of buttery taste and cooling effect attributes of cookies containing xylitol or sucrose during storage. Values for the cookie type with different superscripts (a, b, c) differ significantly (*p* < 0.05).

**Figure 5 foods-12-04270-f005:**
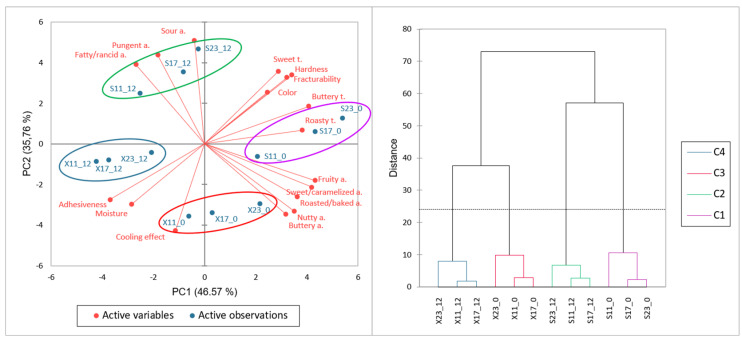
Principal component analysis and cluster analysis dendrogram of sensory attributes of fresh (0) and stored (12) cookies containing 11, 17, and 23% of xylitol (X) or sucrose (S).

**Table 1 foods-12-04270-t001:** Proportions of ingredients in cookie formulations.

Ingredients	Formulations (g/100 g)					
	S11%	S17%	S23%	X11%	X17%	X23%
Refined wheat flour	54.0	48.0	42.0	54.0	48.0	42.0
Sucrose	11.0	17.0	23.0	–	–	–
Xylitol	–	–	–	11.0	17.0	23.0
Butter	26.0	26.0	26.0	26.0	26.0	26.0
Egg yolks	9.0	9.0	9.0	9.0	9.0	9.0

**Table 2 foods-12-04270-t002:** Water activity (a_w_) and pH value of cookies containing xylitol or sucrose (11, 17 & 23%)—fresh and stored for 12 months.

Sweetener Content	Water Activity (a_w_)		pH Value	
	Fresh	Stored	Fresh	Stored
Sucrose				
11%	0.367 ± 0.007 ^aB^	0.398 ± 0.005 ^bB^	5.83 ± 0.07 ^aAB^	5.79 ± 0.14 ^aBC^
17%	0.343 ± 0.007 ^aAB^	0.385 ± 0.008 ^bAB^	5.80 ± 0.08 ^aA^	5.61 ± 0.12 ^aAB^
23%	0.327 ± 0.003 ^aA^	0.378 ± 0.009 ^bA^	5.79 ± 0.11 ^bA^	5.57 ± 0.08 ^aA^
Xylitol				
11%	0.386 ± 0.004 ^aC^	0.394 ± 0.001 ^aAB^	5.95 ± 0.07 ^aBC^	5.93 ± 0.12 ^aC^
17%	0.361 ± 0.009 ^aB^	0.387 ± 0.007 ^bAB^	5.96 ± 0.05 ^aC^	5.90 ± 0.14 ^aC^
23%	0.352 ± 0.005 ^aB^	0.380 ± 0.003 ^bAB^	5.94 ± 0.11 ^aBC^	5.84 ± 0.07 ^aC^

Values with different superscripts in rows (a, b—effect of storage) and columns (A, B, C—effect of sweetener type and content) differ significantly (*p* < 0.05).

**Table 4 foods-12-04270-t004:** Sensory intensity of attributes of cookies containing xylitol or sucrose (11, 17 & 23%)—fresh and stored for 12 months (means ± SD).

Sample	Cookies with Sucrose						Cookies with Xylitol						Xylitol vs. Sucrose	Content, Fresh Cookies	Storage
	Fresh			Stored 12 Months			Fresh			Stored 12 Months					
	11%	17%	23%	11%	17%	23%	11%	17%	23%	11%	17%	23%			
**Appearance**															
Surface colour	4.10 ± 0.22	5.05 ± 0.28	6.89 ± 0.32	3.93 ± 0.24	4.93 ± 0.23	6.97 ± 0.37	2.65 ± 0.18	4.32 ± 0.30	5.80 ± 0.19	2.75 ± 0.34	4.21 ± 0.29	5.73 ± 0.40	*	*	
**Aroma**															
Buttery	5.66 ± 0.22	6.56 ± 0.33	6.75 ± 0.31	2.85 ± 0.33	3.12 ± 0.24	3.37 ± 0.44	5.66 ± 0.24	6.34 ± 0.31	6.20 ± 0.26	4.12 ± 0.35	4.84 ± 0.41	5.10 ± 0.39	*Stored	*	*
Sweet/ Caramelized	3.51 ± 0.27	3.85 ± 0.34	4.29 ± 0.35	1.25 ± 0.32	1.48 ± 0.34	1.64 ± 0.40	2.94 ± 0.33	3.53 ± 0.38	3.71 ± 0.37	1.10 ± 0.22	1.32 ± 0.33	1.52 ± 0.39	*Fresh	*	*
Fruity	1.62 ± 0.43	1.85 ± 0.24	2.10 ± 0.36	0.50 ± 0.15	0.65 ± 0.23	0.78 ± 0.22	1.25 ± 0.27	1.53 ± 0.40	1.85 ± 0.27	0.40 ± 0.19	0.56 ± 0.21	0.64 ± 0.25	*Fresh	*	*
Nutty	2.35 ± 0.32	2.51 ± 0.28	3.04 ± 0.39	0.72 ± 0.21	0.85 ± 0.23	0.92 ± 0.26	2.63 ± 0.48	2.91 ± 0.53	3.32 ± 0.35	0.96 ± 0.19	1.12 ± 0.27	1.24 ± 0.32	*	*	*
Roasted/Baked	4.60 ± 0.23	5.56 ± 0.25	6.27 ± 0.20	2.85 ± 0.34	3.44 ± 0.30	3.82 ± 0.39	4.88 ± 0.31	5.60 ± 0.26	7.31 ± 0.46	3.05 ± 0.36	3.74 ± 0.27	4.13 ± 0.35		*	*
Fatty/Rancid	1.14 ± 0.31	1.42 ± 0.55	1.68 ± 0.43	2.79 ± 0.31	3.10 ± 0.23	3.21 ± 0.33	0.92 ± 0.35	1.24 ± 0.43	1.38 ± 0.44	2.47 ± 0.35	2.70 ± 0.23	2.78 ± 0.32	*Stored		*
Sour	1.42 ± 0.28	1.75 ± 0.31	1.93 ± 0.36	2.52 ± 0.34	2.78 ± 0.21	2.80 ± 0.44	0.85 ± 0.38	1.14 ± 0.33	1.33 ± 0.51	1.42 ± 0.21	1.67 ± 0.39	1.84 ± 0.34	*		*
Pungent	0.41 ± 0.23	0.39 ± 0.18	0.54 ± 0.17	1.85 ± 0.21	2.15 ± 0.24	3.10 ± 0.21	0.39 ± 0.21	0.40 ± 0.16	0.45 ± 0.13	1.01 ± 0.21	1.05 ± 0.18	1.15 ± 0.25	*Stored		*
**Taste**															
Buttery	4.66 ± 0.41	5.96 ± 0.27	6.35 ± 0.29	4.12 ± 0.27	5.17 ± 0.40	5.48 ± 0.24	4.18 ± 0.25	4.84 ± 0.16	5.11 ± 0.23	3.85 ± 0.26	3.87 ± 0.22	4.02 ± 0.36	*	*	*
Sweet	3.54 ± 0.32	5.13 ± 0.41	6.06 ± 0.53	3.59 ± 0.24	5.01 ± 0.39	6.02 ± 0.24	2.21 ± 0.28	3.18 ± 0.21	4.35 ± 0.37	2.16 ± 0.19	3.21 ± 0.21	3.98 ± 0.21	*	*	
Roasty	3.72 ± 0.36	4.50 ± 0.48	4.67 ± 0.38	3.01 ± 0.18	3.34 ± 0.26	3.57 ± 0.27	2.74 ± 0.30	3.08 ± 0.20	4.50 ± 0.45	2.53 ± 0.23	2.76 ± 0.16	4.07 ± 0.35	*	*	*
Bitter	0.18 ± 0.14	0.24 ± 0.13	0.32 ± 0.12	0.32 ± 0.10	0.24 ± 0.08	0.36 ± 0.11	0.19 ± 0.07	0.26 ± 0.09	0.27 ± 0.23	0.41 ± 0.09	0.46 ± 0.13	0.32 ± 0.11			
Aftertaste: cooling effect	0.25 ± 0.14	0.38 ± 0.16	0.36 ± 0.17	0.20 ± 0.11	0.27 ± 0.13	0.31 ± 0.16	2.81 ± 0.17	3.16 ± 0.23	3.89 ± 0.22	1.85 ± 0.19	2.21 ± 0.28	3.10 ± 0.28	*	*Xyl.	*Xyl.
**Texture**															
Hardness	5.07 ± 0.47	6.04 ± 0.48	6.61 ± 0.41	3.82 ± 0.39	4.71 ± 0.27	5.39 ± 0.33	2.43 ± 0.40	2.29 ± 0.34	3.14 ± 0.31	2.37 ± 0.20	2.23 ± 0.25	3.06 ± 0.26	*	*	*Sucr.
Fracturability	6.00 ± 0.53	7.09 ± 0.60	7.59 ± 0.43	4.87 ± 0.36	5.16 ± 0.30	5.95 ± 0.27	3.36 ± 0.29	2.62 ± 0.27	3.02 ± 0.40	3.21 ± 0.28	2.65 ± 0.29	2.93 ± 0.24	*	*Sucr.	*Sucr.
Moisture	2.60 ± 0.47	2.12 ± 0.45	2.01 ± 0.29	3.43 ± 0.42	3.19 ± 0.28	2.85 ± 0.35	4.14 ± 0.41	5.64 ± 0.43	5.63 ± 0.39	5.04 ± 0.29	6.37 ± 0.31	6.49 ± 0.40	*	*	*
Adhesiveness	3.93 ± 0.65	3.00 ± 0.42	3.11 ± 0.32	4.16 ± 0.37	3.59 ± 0.35	4.04 ± 0.29	5.11 ± 0.26	4.80 ± 0.24	4.60 ± 0.37	5.86 ± 0.33	5.69 ± 0.28	5.33 ± 0.38	*	*Sucr.	*

* Significantly different (*p* < 0.05) depending on the type of sweetener (xylitol vs. sucrose), content of sweetener (in fresh cookies), or storage time. *Fresh—Significantly different (*p* < 0.05) only in the case of fresh cookies. *Stored—Significantly different (*p* < 0.05) only in the case of stored cookies. *Xyl.—Significantly different (*p* < 0.05) only in case of cookies containing xylitol; *Sucr.—Significantly different (*p* < 0.05) only in case of cookies containing sucrose.

## Data Availability

Data are contained within the article.

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
