# Peer review of "Comparison of Storage-Related Volatile Profiles and Sensory Properties of Cookies Containing Xylitol or Sucrose"

_foods, 2023, doi:10.3390/foods12234270_

Round 1
Reviewer 1 Report
Comments and Suggestions for Authors
Based on my assessment, the article includes the relevant sections; however, there are a few areas that could benefit from some improvements to enhance professionalism.

Comments on the Quality of English LanguageI want to provide feedback on the quality of the English language used in the paper. While the overall language quality is good, there are a few instances where the use of American English terms could be replaced with their British English equivalents for consistency and clarity.
for example :
In American English, "analyse" is spelled as "analyze".
Additionally, there were a few grammatical errors that need to be corrected.
for example:
consider rewriting the sentence with "were observed" in active voice, e.g. "We observed that".
Reviewer 2 Report
Comments and Suggestions for Authors
Dear authors and editors, thank you for your work and for giving me the chance to read it. I have some recommendations and corrections, which are outlined below. Please review the following comments:
Firstly, the writing style and language of the manuscript need to be revised, as there are numerous grammar mistakes.
The abstract lacks substantial results in the form of numerical data. It should be completely revised to include the following sections: Background and problem, the rationale for the study, research objectives, methodology, important data with statistical analysis, conclusions, novelty, and the importance of the findings.
In the introduction, the authors should include a paragraph discussing volatile compounds, including their identification, their effect on the product, and how they are determined.
What is the novelty of this work?
Please specify the year and month when the flours were purchased, as this is important for determining moisture content.
Provide detailed information about the purity of sucrose and xylitol, as well as all the materials used in the formulations.
Explain the logic behind the formulation ratios, and consider referencing relevant pre-experiments.
The Materials and Methods section lacks detail and references; it needs improvement.
For the RI, the authors should explain the methods they used to prepare the RI solution, how it was injected into the GC-MS, and how it was calculated for the identified volatile compounds.
For the identification and determination of volatile compounds, the authors should consider adding more details. I recommend checking this article with DOI 10.1007/s42770-022-00773-7 for guidance.
Table 2 should be moved to the supplementary materials and not included in the main text.
The statistical analysis needs revision. The authors did not compare the two groups (samples with sucrose or with xylitol).
I suggest the authors include descriptions of the available odors of the volatile components, CAS numbers, IR data, categorization of volatile compounds into groups, and a comparison between the samples.
Provide the GC-MS chromatography data in the supplementary materials.
The discussion section requires improvement with more scientific and logically sound reasoning.
The conclusion could be enhanced by summarizing the major findings.
Comments on the Quality of English LanguageThe writing style and language of the manuscript need to be revised, as there are numerous grammar mistakes.
Reviewer 3 Report
Comments and Suggestions for Authors
Concise, and straight forward work. There are just a few comments to address:
Introduction
Add information about sensory perception of modified cookies or the use of xylitol/sucrose in food. It is a big part of the proposed manuscript, so it should be also integrated here
Material and methods:
For the sensory analysis. Add a reference about sensory evaluation.
Specify if a sensory software or paper were used to capture the data.
Sensory evaluation of fresh and 12 month samples were performed separately. Please, specify.
Data analysis.
Was the panel effect considered? Were sensory results analysed using a 2-way ANOVA considering panelist as random effect?
Results
For results in volatile analysis. Could you also provide the changes in the total peak area?
For sensory results. It would help to include histograms of those significant attributes where you can display the how the different conditions are changing over time.
Reviewer 4 Report
Comments and Suggestions for Authors
There is no explanation in the annotation for choosing xylitol as a sugar replacement. The role of sugar in the formation of the flavor and aroma profiles of cookies has not been disclosed.
It is not clear why, of all the physicochemical indicators, the authors chose pH and water activity. Why were these indicators determined, how are they related to the aroma of cookies.
The article submitted for review contains interesting material regarding the effect of a sugar substitute on the sensory profile of cookies. However, there are a number of comments that may improve the quality of the presented article.
The scale used for descriptive sensory evaluation is poorly described. How applicable is the term spicy to the flavor of cookies?
The overall experimental design is poorly described. Under what conditions were the cookies stored (the authors indicate only the absence of light): temperature, relative humidity? Why was the sensory assessment not performed after 3 months?
Round 2
Reviewer 2 Report
Comments and Suggestions for Authors
no comments
Author Response
Dear Editor,
Thank you so much for acceptation revision of our manuscript.
We are also grateful for having possibility to improve Table which presents volatile compounds in cookies containing xylitol or sucrose (11, 17 & 23%) - fresh and stored for 12 months. Table has number 3 (not 4 we are very sorry for mistake). As Reviewer suggested, Table was supplemented with SD.
Please accept our best regards,
Jarosława Rutkowska